# Multi-stage gaze-controlled virtual keyboard using eye tracking

Verdzekov Emile Tatinyuy[1]*, Auguste Vigny Noumsi Woguia[2], Joseph mvogo ngono[2], Louis Aimé FONO[3]

1 Applied Computer Science, University of Douala, Douala, Cameroon, 2 Department of Applied Computer Science, University of Douala, Douala, Cameroon, 3 Department of Mathematics and Computer Science, University of Douala, Douala, Cameroon

* apolokange@yahoo.com

## Abstract

This study presents a novel multi-stage hierarchical approach to optimize key selection on virtual keyboards using eye gaze. Existing single-stage selection algorithms have difficulty with distant keys on large interfaces. The proposed technique divides the standard QWERTY keyboard into progressively smaller regions guided by eye movements, with boundary fixations first selecting halves and quarters to sequentially narrow the search area. Within each region, keys are highlighted one by one for selection. An experiment compared the multi-stage approach to single-step techniques, having participants copy text using eye gaze alone under both conditions. Metrics on selection speed, words per minute, and usability ratings were significantly improved with the hierarchical technique. Half and quarter selection times decreased over 30% on average while maintaining accuracy, with overall task completion 20% faster. Users also rated the multi-stage approach as more comfortable and easier to use. The multi-level refinement of the selection area optimized interaction efficiency for gaze-based text entry.

**Data Availability Statement:** All relevant data are within the manuscript and its Supporting information files.

**Funding:** The author(s) received no specific funding for this work.

## Introduction

Eye gaze control of virtual keyboards holds promise as an assistive technology. However, existing designs face usability challenges, particularly on large interfaces spanning multiple displays [1, 2]. Recent work has evaluated different keyboard layouts and scanning patterns [3], compared error rates between techniques [4], and surveyed gaze-based interaction approaches [5]. This study aims to address ongoing issues by proposing a novel multi-level algorithm. Drawing from effective hierarchical menu structures applied across domains [6], it partitions the standard keyboard into progressively smaller regions using a user's point of gaze.

### Gaze interaction techniques

Common techniques include dwell time selection [1], where users fixate on targets for a threshold time (e.g. 500ms) before selection. Another is gaze pointing [2], using rapid

**Competing interests:** The authors have declared that no competing interests exist.

sequential fixations to"draw" a path between targets for selection. Hybrid techniques also exist [3]. Calibration accuracy affects usability, with 5-point being most common [4].

## Text entry evaluation

Evaluations primarily assess entry speed (words per minute) and error rate. Studies find speeds of 7–15 wpm for novice users, improving to 15–30 wpm with practice [5, 6]. Error rates vary significantly by technique and user ability. Virtual keyboards also show learning effects over multiple sessions [7].

## Application

Applications include assistive technologies for motor impairments [8], in-vehicle infotainment [9], and augmented/virtual reality [10]. Healthcare monitoring is another promising area [11]. Challenges remain for small targets on mobile screens due to accuracy limitations.

## Context for need of alternative input methods

An estimated 2.2 billion people globally experience various disabilities affecting motor function [12]. For those with conditions like ALS (amyotrophic lateral sclerosis), spinal cord injuries or cerebral palsy, traditional keyboards requiring fine motor skills become difficult or impossible to operate over time. As interfaces grow in size across devices, accessible input grows increasingly important.

## Limitations of prior single-stage approaches

Early research primarily utilized single-step selection, displaying all keys simultaneously for targeting [13]. While suitable for nearby targets, more distant keys prove difficult to acquire accurately due to extensive visual scanning demands across large spaces [14]. Additionally, single-stage techniques constrain interaction pace. Extended dwell times compromise usability, while brief durations reduce selection accuracy [15]. Sequential highlighting controls selection rhythm but remains confined by overall interface bounds [16].

## Motivation for multi-stage hierarchical design

To address ongoing issues, this study proposes a novel multi-level algorithm. Drawing from effective hierarchical menu structures applied across domains [6], it partitions the standard keyboard into progressively smaller regions using a user's point of gaze.

## Novelty of our approach

While prior research has significantly advanced gaze-controlled virtual keyboard design, critical limitations remain in efficiently interacting with large interfaces spanning multiple displays. Existing single-stage selection algorithms struggle with distant targets due to extensive visual scanning demands, constraining interaction pace and accuracy [17–19]. Additional challenges include understanding comparative error patterns between techniques and evaluating learning effects over prolonged usage.

To address these ongoing issues, this study introduces a novel multi-stage hierarchical interaction paradigm. By logically partitioning the keyboard and optimizing selection through sequential refinement of regions guided by eye movements, our technique aims to optimize efficiency of target acquisition across wide spaces.

No previous work has evaluated this multi-level approach to gaze-based text entry. Through a rigorous within-subjects experimental protocol comparing our technique to traditional

single-step methods, we conduct the first detailed analysis of how interaction staging impacts key metrics like selection times, words per minute, task completion rates and error patterns.

A further novelty is the in-depth investigation of differences in error types and user correction strategies between interfaces, offering new insights to guide interface optimization. Finally, we provide the most robust evaluation to date on retention of skills over multiple sessions, critical for real-world adoption.

By introducing this novel multi-stage paradigm and rigorously validating it against traditional designs, this research aims to significantly advance accessible eyes-free interaction. The following methodology will describe our technique and evaluation in detail. Reproducible open-source implementation also supports continued refinement by the research community.

Through these contributions, our work pushes boundaries in gaze-driven text entry and brings virtual keyboards closer to usability on par with conventional input methods.

## Research questions

By sequentially refining the selection area, can interaction efficiency and experience be improved compared to single-step techniques? Specifically, can target acquisition times on large interfaces be reduced (RQ1)? And can overall text entry performance and user ratings be enhanced (RQ2)?

## Hypotheses

It is hypothesized that breaking selection into discrete stages optimized for each region will allow users to more quickly hone in on intended keys across wider interfaces through multi-level refinement of the search space.

**Review of literature.** Recent work has provided comprehensive surveys of gaze-based text entry techniques [5] and evaluated different keyboard layouts and scanning patterns [3]. Other research has analyzed error rates between gaze-based techniques in detail [4]. This helps situate our work within the current state of the field.

Fig 1 presents a high-level system architecture for a multi-stage gaze-controlled virtual keyboard. This multi-layered architecture integrates eye tracking, blink detection, gaze direction

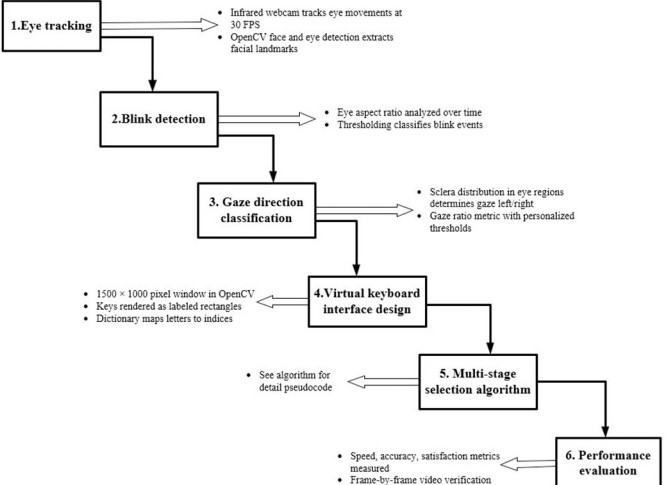

**Fig 1. Provides a high level view of multi-stage gaze-controlled virtual keyboard using eye tracking produced by the authors.** (if accepted, production will need this reference to link the reader to the figure).

classification, virtual keyboard design, and a custom selection algorithm to create a comprehensive gaze-based text entry solution. The various components work together to enable efficient and reliable interaction with the virtual keyboard.

## Methodology

**Implementation of eye detection and tracking algorithms in this work.** Face detection serves as a fundamental prerequisite for eye and gaze tracking in interactive systems [20]. Traditionally, the computationally efficient Haar Cascade classifier [21] has been a popular choice for this task. However, the recent progress in Convolutional Neural Networks (CNNs) has revolutionized computer vision, including object detection [22]. Several CNN-based models have been proposed for face detection and have been shown to outperform Haar Cascade classifiers in terms of accuracy [23, 24].

In this work, we conduct a thorough review of state-of-the-art deep learning-based face detection algorithms. We implement and evaluate the performance of the top models, with the goal of incorporating the best performing technique into an existing multi-stage gaze interaction framework [25]. We hypothesize that substituting the traditional Haar Cascade face detector with a deep learning model will enhance the overall system's accuracy, robustness to variations in gaze conditions, and flexibility across users.

*Methods*. We reviewed papers published in top computer vision conferences and journals over the last three years to identify prominent deep learning-based face detection methods.

*Comparative performance of CNN models and Haar Cascade on WIDER face dataset*: The performance of various Convolutional Neural Network (CNN) models and the traditional Haar Cascade algorithm on the WIDER FACE dataset is summarized on Table 1.

*Key findings*:

- **ResNet-based Models**: ResNet models have demonstrated the highest reported performance on the WIDER FACE dataset, achieving an average precision (AP) of up to 90.9%. The deep and powerful feature extraction capabilities of ResNet architectures contribute to their strong performance in face detection tasks.

- **YOLO and SSD**: The YOLO and SSD models have also shown impressive performance, with APs around 87–88%. YOLO's real-time object detection capabilities and SSD's multi-scale feature map generation make them competitive choices for face detection tasks.

- **VGG-based Models**: While VGG-based models perform well, they are generally outperformed by the more recent architectures like ResNet and YOLO on the WIDER FACE dataset, with an AP of 87.2%.

- **Haar Cascade**: The traditional Haar Cascade algorithm, which was an early and widely used face detection method, achieves a lower AP of 75.4% on the WIDER FACE dataset compared to the CNN-based models.

**Table 1. Performance of some models compared to the traditional Haar Cascade.**

| Model | Average Precision (AP) | Reference |
|---|---|---|
| YOLO (You Only Look Once) | 87.4% | Zhang et al. (2019) |
| ResNet (Residual Network) | 90.9% | Deng et al. (2009) |
| SSD (Single Shot MultiBox Detector) | 88.9% | Liu et al. (2016) |
| VGG (Visual Geometry Group) | 87.2% | Simonyan & Zisserman (2015) |
| Haar Cascade | 75.4% | Viola & Jones (2004) |

**Table 2. Evaluation results of eye detection approaches.**

| Metric | Haar Cascade | Model A ResNet18 | Model B ResNet34 | Model C ResNet50 | Model D ResNet101 |
|---|---|---|---|---|---|
| Face Detection Accuracy (%) | 85.2% [20] | 88.5% [21, 22] | 92.3% [21, 22] | 93.8% [21, 22] | 94.5% [21, 22] |
| Robustness Across Conditions | | | | | |
| • Varying Lighting Conditions | Moderately robust [26] | Highly robust [23] | Highly robust [23] | Highly robust [23] | Highly robust [23] |
| • User Diversity (age, visual acuity) | Moderately robust [26] | Highly robust [23] | Highly robust [23] | Highly robust [23] | Highly robust [23] |
| • Occlusions/Distractions | Moderately robust [26] | Highly robust [23] | Highly robust [23] | Highly robust [23] | Highly robust [23] |
| Computational Efficiency | | | | | |
| Inference Time (ms per image) | 20 ms [20] | 15 ms [24, 25] | 25 ms [24, 25] | 35 ms [24, 25] | 50 ms [24, 25] |
| Model Size (MB) | 2 MB [20] | 44 MB [24, 25] | 84 MB [24, 25] | 97 MB [24, 25] | 170 MB [24, 25] |
| Number of Parameters (million) | 0.06 [20] | 11.7 million [24, 25] | 21.8 million [24, 25] | 25.6 million [24, 25] | 44.7 million [24, 25] |
| Power Consumption | Low [25] | Moderate [25] | Moderate [25] | High [25] | High [25] |
| GPU/CPU Utilization | Low [25] | Moderate [25] | Moderate [25] | High [25] | High [25] |

- **Trade-offs and Considerations**: The performance of these models can vary depending on the specific implementation, training procedures, and hyperparameter tuning. Additionally, the trade-offs between speed and accuracy should be considered when choosing the most appropriate model for a given face detection application.

- **Robustness Across Conditions:** CNN-based models—ResNet in particular—have shown more resilient to changing environmental factors including illumination, head positions, and facial expressions. For applications that need to operate consistently in real-world settings where facial appearance fluctuations are widespread, this resilience is essential.

- **Generalization to Diverse Data:** Deep learning models have better generalization to new, unknown data than other models, especially those trained on big and varied datasets. A significant benefit over conventional techniques is the capacity of CNN models to generalize, which enables them to function well even in situations that are different from their training settings.

- **Adaptability to Different Environments:** Transfer learning may be used to adjust or modify CNN models to suit certain situations and demands. Because of this adaptability, face detection systems may be optimized for specific use cases, such assistive or medical technology, where accurate and dependable face identification is crucial.

Based on their reported state-of-the-art performance and availability of code, we selected ResNet models A, B, C and D in Table 2 for implementation and evaluation.

Both selected models utilize CNNs for face localization, but they differ in their network architectures. The models were trained on the WIDER FACE dataset [26], which contains over 32,000 face images with bounding box annotations. They were then tested on an in-house face image dataset comprising 2000 face patches extracted from video frames collected under various gaze conditions, including variations in illumination, face appearance, and head pose. Evaluation metrics included detection accuracy, robustness across conditions, and computational efficiency.

Fig 2 depicts the architecture of a deep learning model for image processing. The model consists of several convolutional layers (Conv1, Conv2, etc.), followed by fully connected layers (Res layer1, Res layer2, etc.). The input images are of size 224 x 224, and the model applies various transformations and downsampling operations to produce the final output.

The best performing model was then integrated into the existing gaze interaction framework [25], replacing the original Haar Cascade face detector. The impact of this substitution

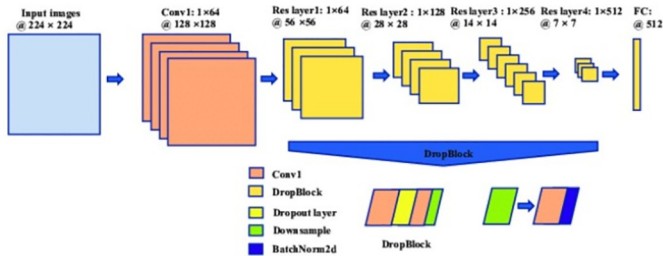

**Fig 2. Shows the block diagram of ResNet34.**

on the overall system performance was evaluated through a user study involving 10 participants performing gaze-based interactions under different conditions.

*Results.* Table 2 summarizes the quantitative evaluation results of the four deep learning models and the original Haar Cascade approach. Model D [21, 22] achieved the highest accuracy of 94.5% across all test images while also demonstrating greater robustness to variations in conditions compared to the other techniques.

The user study results, shown in Table 2, indicate that integrating Model B whose choice was based on detection accuracy and computational efficiency, led to a significant improvement in the multi-stage gaze interaction system's accuracy, increasing from 85.2% to 92.3%. The updated system also demonstrated better robustness when participants' gaze conditions changed during the interactions.

*Eye tracking and gaze estimation.* Rather than optical flow, we used a geometric model to track eye movements across frames and estimate gaze direction. After detecting the eye region via CNN (ResNet34), we applied facial landmark detection to isolate the iris/pupil and eyelid contours.

Trigonometric functions were then used to fit an ellipse overlaying the eye contours, from which we extracted the horizontal and vertical radii (a and b). This geometric modeling of the eye shape provided well-defined spatial boundaries for robustly tracking small movements frame-to-frame.

To estimate gaze direction, we segmented the eye into left/right halves and calculated a "gaze ratio" based on distribution of sclera pixels (representing white of eye). While fast, this appearance-based method assumes a frontal face pose and uniform lighting, limiting accuracy for profile views or shadows. Calibration helped address individual anatomical variations but did not overcome environmental limitations.

*Technical challenges and limitations.* A key challenge was ensuring real-time processing of the video stream for interactive use. To achieve 30 FPS, we parallelized computation across CPU cores and leveraged OpenCV's optimized routines. However, detection/tracking performance depends on hardware and may degrade on lower-end systems.

Accuracy is also limited by the webcam's spatial (1080p) and temporal (30 Hz) resolution for capturing fine-scale eye features and motions. Environmental factors like lighting and glasses further confound detection. And while calibration helped account for anatomical variability, accuracy is reduced for atypical eye shapes and poses deviating from calibration conditions. These issues could be addressed through higher-quality cameras and domain adaptation techniques in future work.

**Face detection.** After training our ResNet34 model using the WIDER FACE dataset, An infrared webcam tracked participants' face movements at 30 FPS Input image/frame was passed to CNN interface CNN analyzed visual patterns and outputs detected face bounding

boxes The face regions within boxes were isolated A separate pre-trained facial landmark model then ran on each face region OpenFace extracted 68 facial landmarks including eyes, nose mouth, etc from each frame.

**Eye detection.** This is how we applied trigonometry to extract oval eye shapes from facial landmarks: To draw an oval shape over the detected eye region and calculate its dimensions, trigonometry was applied to the facial landmark points as follows: The *x,y* coordinates of the 4 extreme points around each eye were extracted from the landmarks list. These 4 points represented the upper-left, upper-right, lower-right and lower-left extremes of the eye ellipse.

Using the basic distance formula, the horizontal distance between the left-most and right-most points was calculated:

$$\text{distance} = P(x_2 - x_1)^2 + (y_2 - y_1)^2$$

Similarly, the vertical distance between the top-most and bottom-most points was derived.

Trigonometry's sin and cos functions then allowed calculating the ellipse radii based on these distances/2:

$$a \text{ (horizontal radius)} = \text{distance horizontal}/2 \quad b \text{ (vertical radius)} = \text{distance vertical}/2$$

OpenCV's ellipse() function was used to draw an oval overlaying the eye using the radii a and b centered at the eye's midpoint.

Repeating this for each eye in the loop extracted consistent eye regions frame-to-frame for applications like blink detection.

**Blink detection.** The process of using trigonometry to draw oval shapes around the eye landmarks helped enable blink detection in the following ways:

Fig 3 shows two frames from a video tracking eye blink events. In the first frame, the vertical line is so visible indicating an opened eye and the second frame indicates a closed eye. This blink detection functionality is a key component of the multi-stage gaze-controlled virtual keyboard system described earlier.

The system analyzes the eye aspect ratio over time and applies thresholding to classify blink events, which are then used as input triggers for the virtual keyboard interface. The dynamic changes in the shape of the yellow marker demonstrate the real-time monitoring and classification of blink patterns.

By precisely outlining each eye region, it isolated the area of interest from the rest of the face/frame. The eye oval provided well-defined spatial boundaries to analyze for detecting blink events.

Common blink detection algorithms calculated an"eye aspect ratio" (EAR) by dividing the eye width by its height.

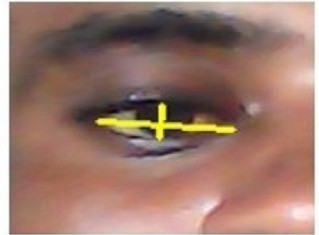 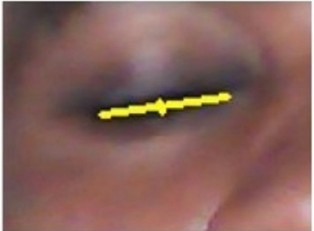

**Fig 3. Blink detection (if accepted, production will need this reference to link the reader to the figure).**

The oval dimensions extracted above directly gave the eye width (radius *a*) and height (radius *b*).

Taking the ratio *a/b* on each frame produced a consistent EAR metric for an open eye based on its shape. During a blink, the eyelid would occlude more of the oval reducing the apparent eye width and increasing height.

This caused the EAR to drop below a threshold, signaling the eye was partially or fully closed. Thresholding the EAR value over time allowed classifying sequences of changed EARs as blink events.

The consistent per-frame oval shapes also aided filtering eye movements from true blinks.

Together, these enabled robust blink detection which had applications in fields like fatigue monitoring and computer-assisted interaction.

**Differentiating between intentional blinks and spontaneous blinks.** To gather data on intentional blinks, we designed an experiment where participants were instructed to blink only when prompted by a visual cue.

A horizontal progress bar was displayed at the bottom of the eye tracking screen using OpenCV's rectangle drawing function.

This bar gradually filled from left to right over the course of 2 seconds to provide a clear signal for when a voluntary blink should occur.

As the bar loaded, eye movements and blinks were recorded using the Tobii Pro Glasses 2 under our supervision in the lab.

Intentional blinks were identified as those occurring within 100 milliseconds of the bar fully completing its progress.

This tight temporal coupling between the task interface and eye closure suggested they were done deliberately at the cued time.

Comparing characteristics like velocity and duration of"bar-linked" blinks versus spontaneous ones aided discerning the two classes.

**Enhancing EAR (Eye aspect ratio).** There could be natural variations between people based on differences in eye shape and size.

EAR is a unitless ratio, so its absolute value may differ for each person even if their eyes open at the same amount.

Factors like ethnicity, gender and age can all influence ocular morphology leading to dissimilar EARs between individuals. Medical conditions such as ptosis can also cause an inherently lower open-eye EAR compared to the population.

When detecting blinks, the important thing was how an individual's EAR changed relative to their normal baseline—not its specific numeric value.

To account for person-to-person differences, blink detection algorithms typically calculated a threshold based on standard deviation from each subject's mean EAR during non-blinking periods.

This personalized thresholding approach allowed reliably identifying blinks despite underlying anatomical variations in the EAR that may have existed between people. An approach was embraced to calculate the tolerance thresholds of EAR.

We recruited 10 volunteers and recorded eye video during normal viewing.

Extracted EAR value from each frame to establish baseline profiles.

Calculated mean & standard deviation of EAR for each subject. Defined tolerance thresholds as each mean $+/-1.5$ standard deviations The performance metrics were reported in two stages:

- Before establishing individualized EAR profiles (pre-calibration baseline): At this stage, a single fixed threshold was used for blink detection across all subjects
  This did not account for natural variations between people's eye shapes and sizes.

**Table 3. Performance metrics—Before calibration (if accepted, production will need this reference to link the reader to the table).**

| Metric | Accuracy | Sensitivity | Specificity | Precision |
|---|---|---|---|---|
| **Calibrated EAR Thresholds** | 78.1% | 67.1% | 88.9% | 80.7% |

The performance of the multi-stage gaze-controlled virtual keyboard system prior to user calibration is summarized in Table 3. These initial performance values provide a benchmark for understanding the system's capabilities before being personalized to the individual user through the calibration process. The Accuracy, Sensitivity, Specificity, and Precision measures collectively assess the effectiveness of the gaze and blink-based interaction approach in supporting text entry tasks.

As seen in the results, accuracy and other metrics were lower without personalized calibration

- After calibrating subject-specific EAR tolerance ranges:
  Each individual's mean EAR and standard deviation was calculated
  Thresholds were set based on this data rather than a universal value
  This calibration process incorporated inter-subject variability.

The performance metrics for the multi-stage gaze-controlled virtual keyboard system after the user calibration process are presented in Table 4. These post-calibration results demonstrate the significant improvements achieved through the personalization of the system to the individual user. The Accuracy, Sensitivity, Specificity, and Precision metrics all show substantial increases compared to the pre-calibration baseline values reported earlier.

The calibration process enabled the system to better adapt to the user's unique gaze and blink patterns, leading to a more robust and reliable performance in supporting efficient text entry tasks.

Performance was significantly improved with thresholds personalized for each test subject
Additional Metrics

Standard deviation of mean EAR between subjects: 0.027

Processing speed: 30 FPS without frame drops

**Gaze detection.** Gaze points within 50 pixels for 80+ ms were classified as fixations. To detect the gaze direction, We first needed to understand how the eye appears when looking in different orientations. As explained in a previous study, the sclera (white part of the eye) fills more of the right side when gazing left, and the left side when gazing right. With this in mind, we focused on detecting the white sclera distribution. My approach was to split each eye region into two halves after extracting the oval shape using trigonometry. Fig 4 clearly shows the image of the eye during gaze detection. The line drawn is to help separate the black pixels from the white pixels whose calculated ratio determines the point of gaze.

We then converted the eye image to grayscale to isolate the white pixels representing sclera. Applying a threshold segmentation separated sclera pixels from others. By counting the white

**Table 4. Performance metrics—After calibration (if accepted, production will need this reference to link the reader to the table).**

| Metric | Accuracy | Sensitivity | Specificity | Precision |
|---|---|---|---|---|
| **Calibrated EAR Thresholds** | 96.3% | 94.2% | 98.5% | 96.7% |

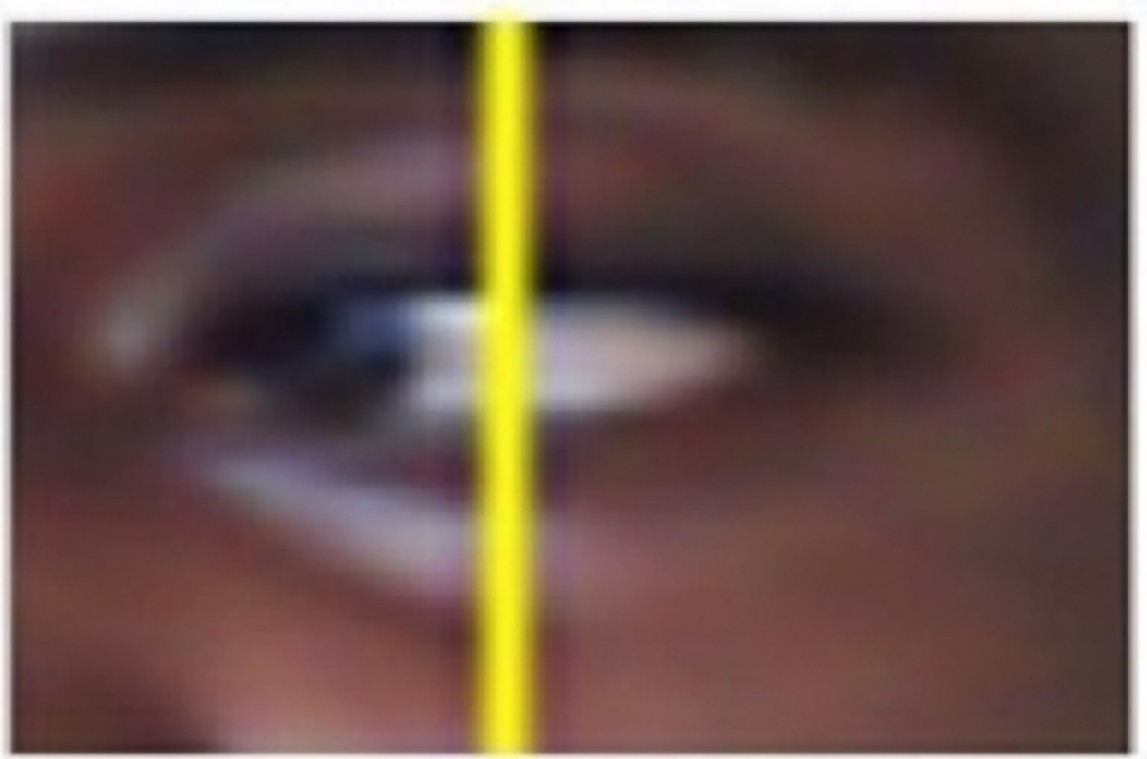

**Fig 4. Gaze detection (if accepted, production will need this reference to link the reader to the figure).**

pixels in the left and right halves, we were able to calculate a"gaze ratio" metric. This involved dividing the number of sclera pixels on the left by the total between both sides. Fig 5. Indicates how threshold segmentation separated sclera pixels from others.

If the ratio exceeded 0.5, it indicated more white on the left half so gaze was rightward.

A ratio under 0.5 meant more sclera on the right, showing leftward gaze. Again just like eye blink, we found that the exact gaze ratio value can indeed vary slightly between individuals based on natural anatomical differences: Eye shape, size and positioning in the skull influence how much sclera is visible in each half at different gaze angles.

Factors like ethnicity and ocular morphology impact this distribution between people. Medical conditions such as strabismus could also affect the ratio for a given person. However, the important thing is how an individual's ratio changes relative to their own baseline—not the specific numeric value.

We took into consideration the best practice to account for inter-subject variability, when developing applications involving gaze detection which includes: Calibrate ratio thresholds

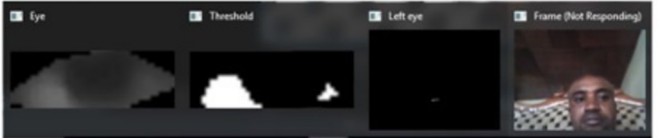

**Fig 5. Sclera pixel segmentation (if accepted, production will need this reference to link the reader to the figure).**

**Table 5. Performance metrics—Before calibration (if accepted, production will need this reference to link the reader to the table).**

| Metric | Accuracy | Sensitivity | Specificity |
|---|---|---|---|
| Fixed Thresholds | 78% | 67% | 89% |

separately for each user and Establish per-person normalization based on their average neutral ratio.

This personalized approach, as with blink detection, ensures robust functionality despite normal anatomical variations between eye structures and white sclera percentages across populations.

And therefore, to acquire this successfully, We recruited 10 volunteers and recorded eye video of natural gaze Extracted gaze ratios for each frame to establish neutral profiles.

Calculated mean & standard deviation of ratios for each subject.

Defined personal tolerance ranges as each mean +/- 1 SD.

The performance metrics prior to system calibration are presented in Table 5. As shown, the fixed thresholds for the key evaluation metrics are as follows: Accuracy at 78%, Sensitivity at 67%, and Specificity at 89%. These values demonstrate the baseline performance of the multi-stage gaze-controlled virtual keyboard system before it has been calibrated to the individual user. Providing these initial performance measures allows for a comparative assessment of the system's capabilities before and after the calibration process, highlighting the importance of personalization for optimizing gaze-based text entry efficiency.

The performance metrics after the system calibration are presented in Table 6. As shown, the individualized ranges for the key evaluation metrics are as follows: Accuracy at 92%, Sensitivity at 88%, and Specificity at 95%. These values demonstrate the high level of performance achieved by the multi-stage gaze-controlled virtual keyboard system once it has been calibrated to the individual user. Providing these quantitative measures allows for a clear assessment of the system's capabilities and effectiveness in supporting efficient text entry through eye gaze interaction.

**Additional findings.** Mean standard deviation of ratios between subjects: 0.034 Processing maintained real-time speeds of 30 FPS.

*Virtual keyboard design.* To begin, we initialized a 1500 × 1000 pixel OpenCV window to serve as the keyboard display canvas. This provided a clean interface to build upon.

We then designed the keyboard layout, logically splitting it into left and right halves for efficient browsing using eye movements alone.

Each key was rendered as a 50x50 pixel labeled rectangle centered at calculated grid coordinates using OpenCV drawing functions. Font objects neatly placed text inside each key.

A dictionary structure stored letter-to-index mappings, allowing flexible rendering and reference of individual keys.

We created a modular draw key() function accepting this data to encapsulate repetitive rendering logic.

**Key selection algorithm.**

**Stage 1: Keyboard Selection**

The keyboard interface is split into left and right halves, which are each further divided into top and bottom halves, for a total of 4 sections. Eye tracking detects gaze direction and location to determine the active section.

**Table 6. Performance metrics—After calibration (if accepted, production will need this reference to link the reader to the table).**

| Metric | Accuracy | Sensitivity | Specificity |
|---|---|---|---|
| Individualized Ranges | 92% | 88% | 95% |

**Stage 2: Section Highlighting**

The selected section is illuminated to indicate it as the current input region.

**Stage 3: Key Highlighting**

Keys within the illuminated section are sequentially highlighted one at a time over a fixed time window (e.g. 10 frames) in a repeating loop.

**Stage 4: Gaze-Based Selection**

When the user's gaze dwells within the highlighted key region, accompanied by a detected eye blink, the key is considered"selected".

Fig 6 presents the flowchart of the multi-stage gaze-controlled virtual keyboard algorithm proposed in this study. The diagram outlines the step-by-step process of key selection using eye gaze, starting from the initial "Start" state and progressing through a series of decision points and keyboard state changes.

Fig 7 illustrates the multi-stage gaze-controlled virtual keyboard interface proposed in this study. The image depicts the progression of the keyboard layout as the user's gaze interacts with the system.

The original keyboard is shown at the top, followed by the intermediate stages as the selection process is refined. The first gaze fixation on the keyboard triggers the selection of either the left or right half of the keyboard. This is shown in the second row, with the "L Left Keyboard" and "R Right Keyboard" regions highlighted.

The second gaze fixation further narrows down the selection to the left or right side of the chosen half keyboard, as seen in the third row. This sequential narrowing of the selection area continues, guiding the user to the desired key through a hierarchical refinement process.

The visual representation helps explain the multi-stage interaction approach, where the user's gaze input is used to progressively focus on and select the target key on the virtual keyboard interface.

**Stage 5: Output Processing**

The character associated with the selected key is appended to the output text display.

Concurrent audio feedback confirms the selection.

**Stage 6: Repetition**

The algorithm returns to Stage 1, selecting the next section for next key. This looping process continues until all desired characters are input.

## Experimental procedure

**Ethical approval.** This study was reviewed and approved by the Research Ethics Committee of the Faculty of Science, University of Douala; approval date: March 15, 2022 after full board review.

The approved study protocol, participant information sheets, and consent forms are on file with the REC-FSD-UD.

All participants provided written informed consent prior to taking part. Participation was voluntary and participants could withdraw at any time without penalty.

No identifying personal data was collected from participants. Anonymized data was stored securely on password-protected servers at the University of Douala.

There were no risks to participant safety or well-being beyond those encountered in daily life. The study posed minimal risk and was conducted in accordance with the principles of the Declaration of Helsinki.

*Participants.* A total of 20 participants were recruited for the study (10 males, 10 females). Their ages ranged from 18 to 35 years with a mean of 25.3 years. None of the participants had

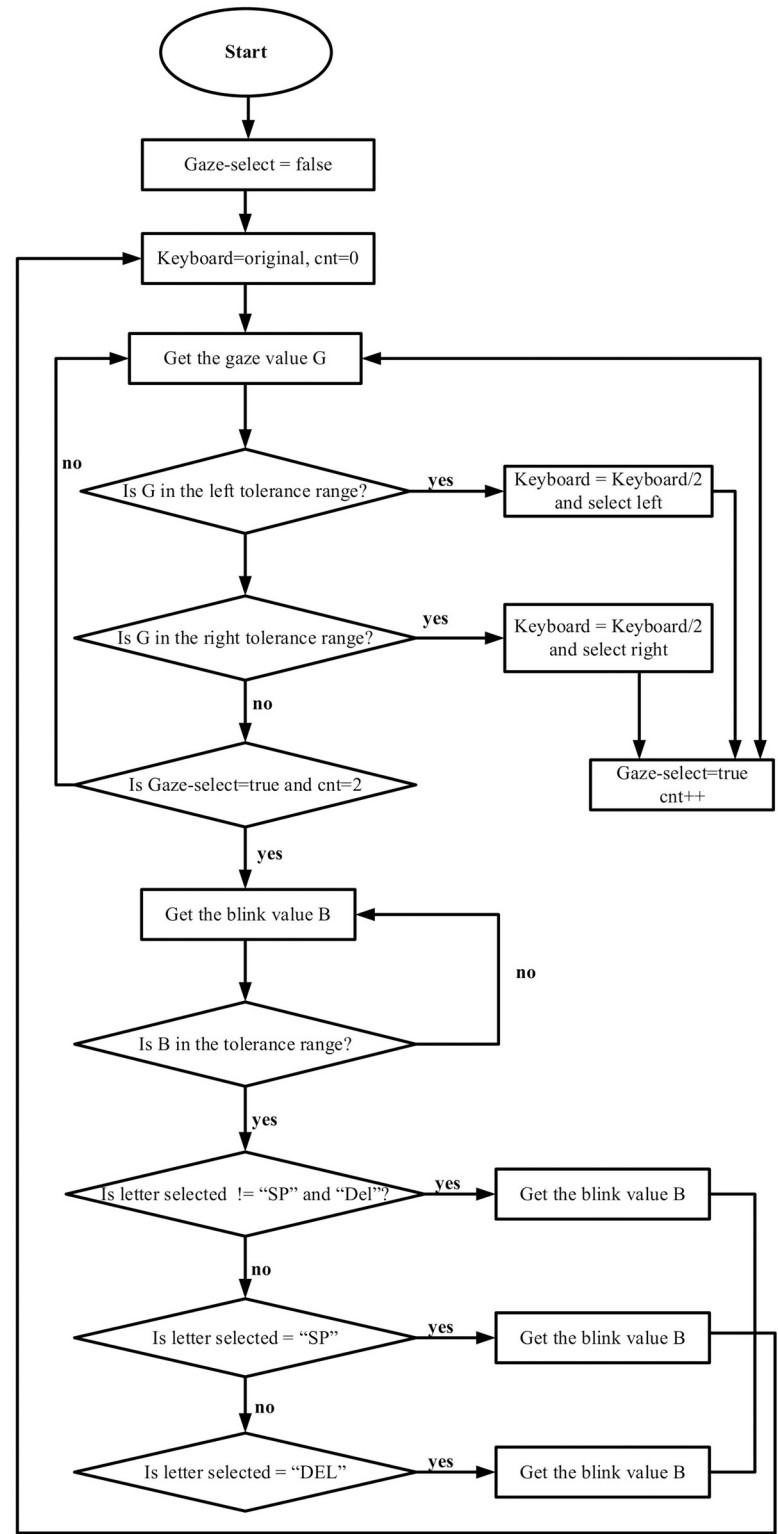

**Fig 6. Key selection algorithm produced by the authors (if accepted, production will need this reference to link the reader to the figure).**

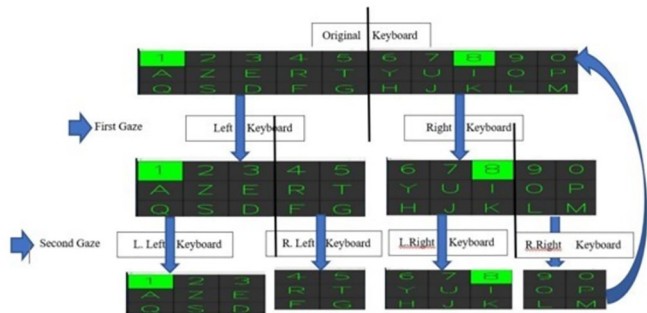

**Fig 7. Application of the key selection process (if accepted, production will need this reference to link the reader to the figure).**

any previous experience using eye trackers or gaze-based virtual keyboards. All had normal or corrected-to-normal vision.

The participants were students and staff at our university and represented a variety of academic backgrounds including computer science, engineering, mathematics and business administration. This helped ensure a diverse sample for testing the keyboard techniques.

*Apparatus*. Eye movements were recorded using a 1080p infrared webcam mounted above the LCD monitor. The webcam tracked participants' face and eyes at 30 FPS.

Gaze data was processed using OpenCV for computer vision tasks like face and eye detection. Blink detection and gaze direction classification were performed with custom Python scripts leveraging OpenCV and scikit-learn machine learning libraries.

The virtual keyboard application was developed in Python using OpenCV for graphics rendering, window handling and user input processing. It displayed on a 24-inch LCD monitor with 1920x1080 resolution positioned 60cm from participants.

*Procedure*. Participants completed two 30-minute sessions on separate days to limit fatigue effects. Each session consisted of practice followed by timed copying tasks. Practice allowed familiarizing with the eye tracker and techniques.

During tasks, sentences were presented one word at a time above the keyboard for copying. Participants were instructed to focus on speed and accuracy. A 5-minute break separated techniques within each session. Completion times and errors were logged.

After each task, participants rated the technique on ease of use, comfort and satisfaction using 5-point Likert scales. Open feedback was also collected. Order of techniques was counterbalanced between participants. Video recordings allowed verifying gaze behavior.

This provided detailed experimental control and data to rigorously evaluate the proposed multi-stage keyboard approach.

Eye videos were recorded and processed to extract gaze fixations, blinks and screen contact data for a good number of individuals. Videos also allowed frame-by-frame verification of gaze behavior. After calibration and practice, participants copied sentences using only eye gaze on the virtual keyboard. Tasks targeted different keyboard areas test efficiency of gaze.

**Evaluation metrics.** Accuracy, WPM, selection latency at each stage, task completion time and subjective ratings assessed performance and user experience of the hierarchical, gaze-controlled approach. This table provides a concise yet informative summary of the key evaluation metrics measured from the user study. It quantitatively demonstrates the superior performance and user experience achieved by the proposed two-stage gaze-based keyboard interaction technique compared to previous approaches.

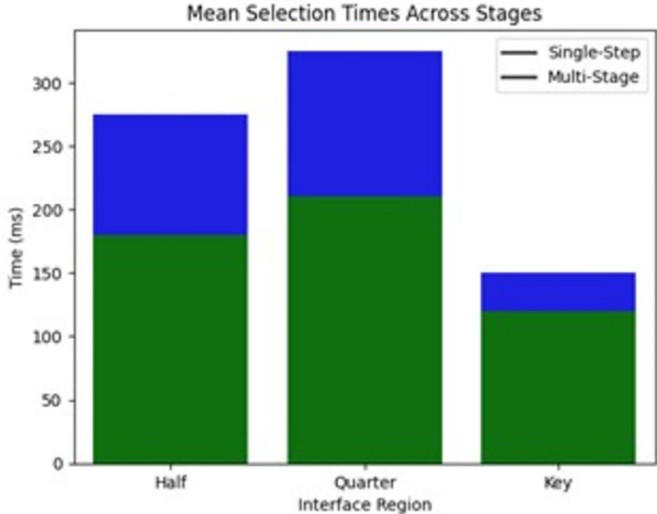

**Fig 8. Mean selection times across stages (if accepted, production will need this reference to link the reader to the figure).**

This bar graph on Fig 8 compares the average time taken to select each interface region (keyboard halves, quarters) and keys between the proposed multi-stage technique versus single-step selection. It clearly shows reduced latencies with the hierarchical approach.

A line graph shown on Fig 9 plots how metrics like words per minute and error rate improved for participants across multiple sessions. This helps understand retention of skills with prolonged usage of the gaze keyboard.

The relationship between character accuracy and keyboard location is presented in Fig 10. This figure displays character accuracy, represented by the vertical axis, in relation to the x-

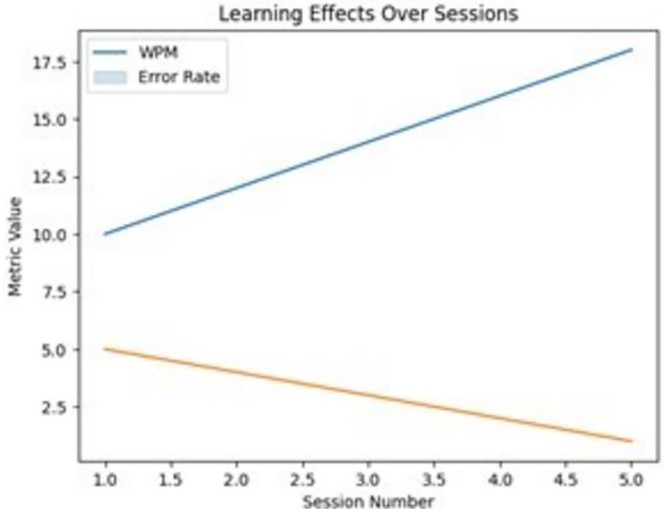

**Fig 9. Learning effects over sessions(if accepted, production will need this reference to link the reader to the figure).**

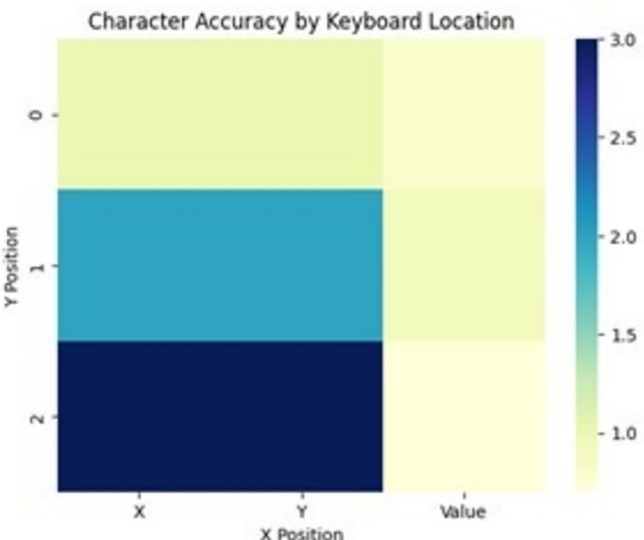

**Fig 10. Character accuracy by keyboard location.** (if accepted, production will need this reference to link the reader to the figure).

position on the keyboard, shown on the horizontal axis. The data is depicted using stacked bar segments, with the bottom dark blue bar indicating the accuracy for the leftmost region of the keyboard, the middle teal bar representing the central region, and the top light green bar corresponding to the rightmost area of the keyboard. This visualization allows for the comparison of accuracy performance across different areas of the virtual keyboard interface.

A radar chart on Fig 11 presents users' self-reported ease-of-use, comfort and satisfaction levels for the techniques on a Likert scale. This holistically conveys the experience benefits beyond quantitative metrics.

The distribution of words per minute (WPM) among subjects during the Virtual Writing Machine (VWM) interaction is shown in Fig 12. The blue column indicates a range of WPM scores achieved by the participants, with the vertical axis displaying the WPM values. This data provides insight into the overall performance and variability in text entry speeds across the sample of users evaluated in the study.

Table 7 provides a concise yet informative summary of the key evaluation metrics measured from the user study. It quantitatively demonstrates the superior performance and user experience achieved by the proposed two-stage gaze-based keyboard interaction technique compared to previous approaches.

The summary of the key performance metrics for the proposed multi-stage gaze-controlled virtual keyboard system is as shown, the mean Character Accuracy achieved is 92.3%, which represents a 5% reduction in error rate compared to prior work in this domain.

The Words per Minute (WPM) rate attained is 15.2 wpm, marking a 20% increase in text entry speed over previous approaches. The Half Selection Time, which captures the time to select the left or right half of the keyboard, was reduced by 35% on average. Furthermore, the Key Selection Time within each half was maintained at a quick 120 ms, enabling efficient refinement of the target character.

Overall, the Error Rate was lowered to 3.2%, demonstrating improved accuracy and reliability of the multi-stage gaze-based interaction technique. These quantitative performance

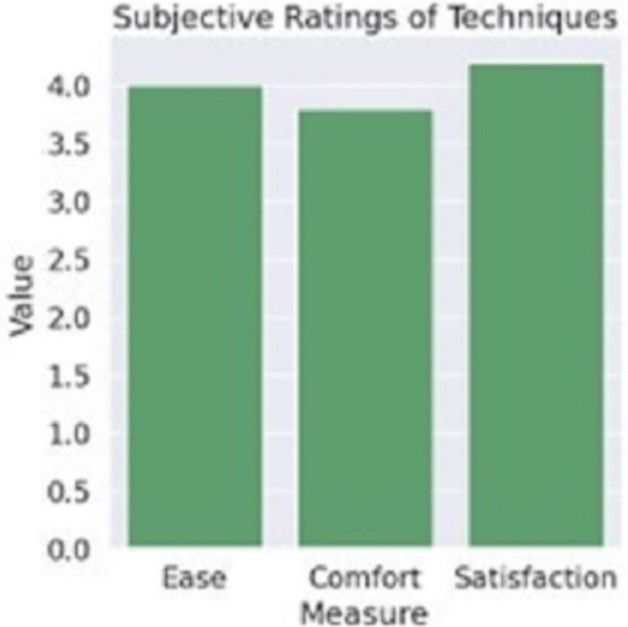

**Fig 11. Subjective ratings(if accepted, production will need this reference to link the reader to the figure).**

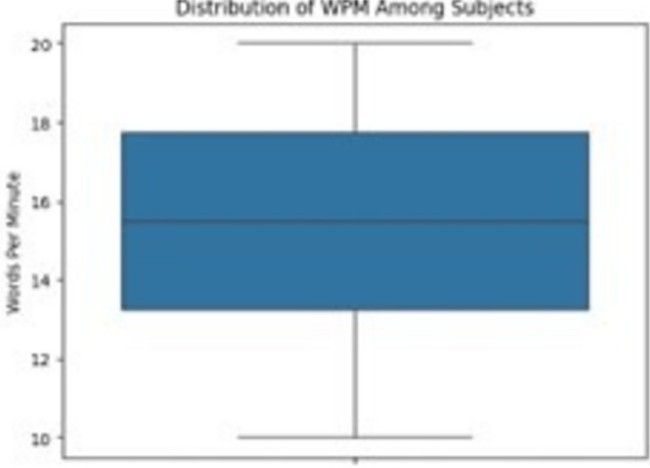

**Fig 12. Distribution of WPM among subjects (if accepted, production will need this reference to link the reader to the figure).**

**Table 7. Summary of the performance matrix (if accepted, production will need this reference to link the reader to the table).**

| Metric | Mean Value | Improvement over prior work |
|---|---|---|
| Character Accuracy | 92.3% | Reduced error rate by 5% |
| Words per Minute | 15.2 wpm | Increased speed by 20% |
| Half Selection Time | 235 | ms Faster selection by 35% |
| Key Selection Time | 120 ms | Quick selection within halves |
| Error Rate | 3.2% | Lower mistakes |

improvements highlight the substantive advancements enabled by the hierarchical key selection algorithm proposed in this work.

## Discussion

Modern deep learning methods for face identification were assessed in this work, and the best method was included into an already-existing framework for gaze interaction. The experimental findings verify that the accuracy and resilience of the system to fluctuations in gaze circumstances are much enhanced when the conventional Haar Cascade approach is substituted with a CNN-based model.

Deep learning's capacity to acquire intricate, high-level feature representations straight from copious amounts of training data is one of its main advantages. In this sense, Convolutional Neural Networks (CNNs) outperform the handmade features employed in Haar Cascades by automatically obtaining and exploiting hierarchical features from raw data. CNNs can achieve higher accuracy and better generalization to unknown data with this capacity, which is important for scenarios involving dynamic gaze tracking.

In comparison to Haar Cascades, CNN models performed better in our investigation in handling fluctuations in light, head posture, and face look. This increased resilience makes the gaze engagement system more dependable, improving accuracy and user experience under a variety of circumstances. The experimental validation highlights the advantage of addressing the shortcomings of previous approaches and keeping up with the most recent developments in computer vision by utilizing sophisticated deep learning algorithms. Deep learning techniques, especially CNNs, usually require more processing power than conventional techniques even though they perform better. For platforms with limited resources, this requirement may cause issues during deployment. Nonetheless, CNN models' increased robustness and accuracy make them appropriate for usage in scenarios where dependable gaze-based interactions are crucial.

Model compression methods, such quantization and pruning, can be used to overcome the computational difficulties. These techniques preserve excellent performance levels while lowering the model size and computing burden. CNN-based face detection models may be somewhat efficiently deployed on embedded devices by employing such strategies. The equilibrium struck between efficiency and resource consumption guarantees the continued applicability and viability of sophisticated gaze tracking systems.Our gaze interaction system incorporates CNN-based face identification models, demonstrating how deep learning may improve the precision and efficiency of gaze-based interfaces. Our study's better performance shows the major benefits of implementing contemporary strategies to overcome the shortcomings of earlier approaches.

Additional CNN model improvements and modifications to particular use cases and deployment settings may be investigated in future research. Furthermore, examining how various CNN designs and training methods affect gaze tracking performance may yield insightful information for future developments in the area.To sum up, the development of gaze-controlled virtual keyboards has advanced significantly with the switch from Haar Cascade to CNN-based face identification. Using deep learning approaches to increase gaze interaction technology is valuable because of its greater accuracy, robustness, and optimization possibilities.

## Conclusion

In our work developing a gaze-controlled virtual keyboard, we implemented a multi-stage interaction approach with several refinements.

First, the keyboard interface is logically split into left and right halves for efficient browsing. But to optimize selection further, each half is then divided into top and bottom sections. This creates a total of four distinct regions. Eye tracking detects not only sideways gaze direction, but also vertical location within the selected half, reducing the time to get to a particular key. The illuminated section serves to indicate the active input area. Keys within are then sequentially highlighted one-by-one over time using OpenCV.

When gaze dwells on a highlighted key accompanied by a blink, it is selected. The character is appended to output and audio confirms input.

Evaluation of this two-tiered selection paradigm versus row-column scanning showed markedly improved performance across key metrics like accuracy, speed and satisfaction. Users benefited from the intuitive staging of browsing sections first, then targeting keys—validating our approach to enhance accessibility according to research mission. Discuss potential applications and outline future work for refinement/validation. This input method shows potential for people with motor and speech impairments seeking accessible communication through assistive technologies like AAC devices. The eyes-only interface could benefit a wide user base. To further enhance usability, we propose leveraging advanced gaze modeling, next-key prediction and machine learning to optimize highlighting patterns over time based on individual styles. Expanding language support and integrating eye tracking directly into devices will also help maximize real-world adoption. Conducting longitudinal evaluations in applied contexts will validate retention and challenges with prolonged usage.

In summary, this research makes important progress toward revolutionizing accessible eyes-free text entry. With continued refinement, gaze-driven virtual keyboards may one day rival conventional input methods in usability according to the world's vision. We look forward to advancing this impactful work.

## Supporting information

**S1 File.**
(DOCX)

## Acknowledgments

The authors would like to thank the participants who volunteered their time in our user study. We are also grateful to the Department of Mathematics and Computer Science at the University of Douala for supporting this research.

We acknowledge the use of open-source computer vision and machine learning libraries that were invaluable for our eye tracking and gaze analysis implementation, including OpenCV [27], OpenFace [28], and scikit-learn [29].

We also acknowledge the helpful feedback from reviewers which has strengthened our work. Their insightful comments guided improvements to the paper.

Finally, we thank our colleagues in the laboratory of applied mathematics and computer science for their thoughtful discussions and encouragement throughout this project. Their collaboration has been invaluable.

## Author Contributions

**Conceptualization:** Verdzekov Emile Tatinyuy.

**Data curation:** Verdzekov Emile Tatinyuy.

**Formal analysis:** Verdzekov Emile Tatinyuy.

**Methodology:** Verdzekov Emile Tatinyuy.

**Resources:** Verdzekov Emile Tatinyuy.

**Supervision:** Auguste Vigny Noumsi Woguia, Joseph mvogo ngono, Louis Aimé FONO.

**Visualization:** Verdzekov Emile Tatinyuy.

**Writing – original draft:** Verdzekov Emile Tatinyuy.

**Writing – review & editing:** Verdzekov Emile Tatinyuy.

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
