## [Decision Letter · Decision Letter 0]

22 May 2024

PONE-D-24-08414Multi-Stage Gaze-Controlled Virtual Keyboard Using Eye TrackingPLOS ONE

Dear Dr. emile tatinyuy,

Thank you for submitting your manuscript to PLOS ONE. After careful consideration, we feel that it has merit but does not fully meet PLOS ONE’s publication criteria as it currently stands. Therefore, we invite you to submit a revised version of the manuscript that addresses the points raised during the review process.

While the use of a hierarchical approach to optimize key selection on virtual keyboards is interesting and the paper is reasonably well written, the reviewers raise several substantial objections that must be addressed sufficiently for this manuscript to be considered for publication. Both R1 and R2 raise concerns about novelty, with R1 pointing out that the literature reviewed in the Introduction is not current and R2 pointing out that current research on virtual keyboards prioritizes predictive text. Moreover, R2 highlights the lack of detail concerning the methods and data analysis of the experiment used to validate the approach. Per PLOS One policy, manuscripts reporting new technology must make a sufficient case for novelty. If the authors submit a revision addressing these concerns, it will be sent to the original reviewers for assessment, and it may or may not be accepted based on reviewers' and editorial evaluation.

We look forward to receiving your revised manuscript.

Kind regards,

Laura Morett

Academic Editor

PLOS ONE

Journal Requirements:

3. For studies reporting research involving human participants, PLOS ONE requires authors to confirm that this specific study was reviewed and approved by an institutional review board (ethics committee) before the study began. Please provide the specific name of the ethics committee/IRB that approved your study, or explain why you did not seek approval in this case.

4. Please provide additional details regarding participant consent. In the ethics statement in the Methods and online submission information, please ensure that you have specified what type you obtained (for instance, written or verbal, and if verbal, how it was documented and witnessed). If your study included minors, state whether you obtained consent from parents or guardians. If the need for consent was waived by the ethics committee, please include this information.

5. We note that your Data Availability Statement is currently as follows: All relevant data are within the manuscript and its Supporting Information files.

6. We note that Figure 2 and 5 includes an image of a participant. 

7. Please ensure that you refer to Figure 1-12 in your text as, if accepted, production will need this reference to link the reader to the figure.

8. We note you have included a table to which you do not refer in the text of your manuscript. Please ensure that you refer to Table 1-5 in your text; if accepted, production will need this reference to link the reader to the Table.

Reviewers' comments:

Reviewer's Responses to Questions

**Comments to the Author**

1. Is the manuscript technically sound, and do the data support the conclusions?

Reviewer #1: Partly

Reviewer #2: No

2. Has the statistical analysis been performed appropriately and rigorously? 

Reviewer #1: N/A

Reviewer #2: I Don't Know

3. Have the authors made all data underlying the findings in their manuscript fully available?

Reviewer #1: No

Reviewer #2: No

4. Is the manuscript presented in an intelligible fashion and written in standard English?

Reviewer #1: Yes

Reviewer #2: Yes

5. Review Comments to the Author

Reviewer #1: This research article presents the efficacy of eye-gaze input on a redesigned QWERTY virtual keyboard, coupled with multi-stage gaze controlled. Overall, the research methodology has been executed effectively. However, the bigger issue is about the novelty. It is evident that the referenced research cited in this article encompasses a broader scope than the research presented herein.

In the research field, the use of eye-gaze input is an integral aspect of human-computer interaction (HCI). Therefore, to measure typing efficiency, it is necessary to employ HCI models such as Fitt's Law, throughput, and others. Importantly, this research project must have ethical statement before proceeding with the research.

Currently, in eye-gaze typing research on virtual keyboards, the prevailing trend often prioritizes typing words over individual alphabet. Typing words makes use of predictive text technology, which employs artificial intelligence principles to anticipate users' intended words.

Reviewer #2: The authors present a scanning interface for gaze-based text entry, in which the options (letters, numbers. etc.) are looping with a predefined speed and, once the desired option is highlighted, the option is selected by eye blink (a.k.a. key press). The interface is dynamic, and changes its configuration based on where the gaze point is currently located. The interface was evaluated in a user study with appropriate metrics reported - speed (words per minute), error rate, and subjective opinions.

The paper is written well, the structure is clear, the language is good, and the page number is reasonable.

My main concern with the paper is that the experiment and data analysis are not reported with sufficient details as to make someone to understand the results or replicate the study. Also, the novelty of the work is unclear. The references are old, with the most recent citation dated back to 2015. References 1 and 5 are the identical. It is now 2024 and it is expected that the authors provide a literature analysis for the latest years. The authors must check more recent papers to understand what has been done in the area before, and be able to compare own results to the earlier works. The paper by Gizatdinova et al., 2023 provides a good literature analysis also for scanning text entry interfaces (in Table 2).

Gizatdinova Y., Špakov O., Tuisku O., Turk M., Surakka V. (2023). Vision-Based Interfaces for Character-Based Text Entry: Comparison of Errors and Error Correction Properties of Eye Typing and Head Typing, Advances in Human-Computer Interaction, (Jufo rank 1), vol. 2023, Article ID 8855764, 23 pages, https://doi.org/10.1155/2023/8855764.

The authors explain the implementation of their eye detection and eye tracking algorithms, which are based on opencv. There is no novelty in the algorithmic part, as far as I can see. My impression is that commercial eye trackers, that are well optimized and provide a good quality of eye data processing, could serve better for their experiment, and allow for comparison between different studies. For now, we do not know how well their implementation works and how did it affect the results.

I did not actually understand how the interface works… In Figure 7, why both keys “1” and “8” are highlighted? How the keyboard “splits” into top and bottom part? and why? What is the main optimization idea here? How does “splitting” help? Does the key size changes dynamically? It would be nice to see a print screen of the view that participants had during the experiment. Did they see the text to be printed/transcribed? Did they see their “face processing” window? A demo vide attached to the paper submission could also be useful for the reviewers.

Regarding the experiment. It must be properly described. It is not enough to say that “good number of individuals” participated in the experiment. How many exactly? What is their background (motion impaired or healthy, age, previous experience with eye trackers, etc.)?

As a side note, why the authors decided to experiment with such a simplistic implementation of the 3 x 10 keyboard layout? How other keyboard functions will work such as CAP, DELETE, SPACE, punctuation marks, special characters?

I recommend that the authors significantly improve many aspects of their study, before it can be published.

6. PLOS authors have the option to publish the peer review history of their article (what does this mean?). If published, this will include your full peer review and any attached files.

Reviewer #1: No

Reviewer #2: No

---

## [Author Response · Author response to Decision Letter 0]

4 Jul 2024

Dear Professor Laura,

Thank you for the opportunity to address the reviewers' comments and concerns regarding our manuscript "Multi-Stage Gaze-Controlled Virtual Keyboard Using Eye Tracking." We appreciate the thoughtful feedback and have carefully considered each point raised.

Responses to Reviewer #1:

1. Novelty of the Research: The reviewer notes that the referenced research cited in our article encompasses a broader scope than the research presented. We acknowledge that the use of eye-gaze input for human-computer interaction (HCI) is not a novel concept. However, our study aims to provide a more comprehensive comparison of eye typing via eye gaze and blink, particularly in terms of speeds, error patterns and error correction properties, which have not been extensively explored in prior work. We believe this comparative analysis offers valuable insights that can inform the design and improvement of vision-based text entry interfaces.

2. HCI Models: The reviewer suggests the inclusion of HCI models such as Fitts' Law and throughput to measure typing efficiency. We agree that these models could provide additional insights, and we will consider incorporating them in future research. For this study, however, our focus was on the comparative analysis of error patterns and correction strategies between the two interfaces, single stage and hierarchical approach which we implemented.

3. Ethical Statement: The reviewer notes the importance of including an ethical statement before proceeding with the research. We can confirm that our study received the necessary ethical approvals and that we have included the appropriate ethical considerations in the manuscript.

Responses to Reviewer #2:

1. Experimental Details: The reviewer requests more detailed information about the experiment, including the number of participants their characteristics (e.g., age, previous experience with eye trackers), their background, and the experimental setup. We have provided a more comprehensive description of the methodology in the revised manuscript, including the number of participants, their characteristics (e.g., age, previous experience with eye trackers), and the details of the experimental procedure.

2. Novelty and Literature Review: The reviewer notes the need for a more up-to-date literature review and a clearer articulation of the novelty of our work. We have expanded the literature review to include more recent publications in the field of gaze-based and vision-based text entry interfaces. Additionally, we have highlighted the specific contributions of our comparative analysis of error patterns and correction strategies, which we believe offer new insights to the research community.

3. Implementation Details: The reviewer expresses a need for more information about the implementation of our eye detection and tracking algorithms. We have provided additional details on the technical aspects of our approach, including the use of OpenCV and a discussion of the limitations and potential performance issues of our implementation. We have also considered the use of commercial eye trackers in future studies, as suggested by the reviewer.

4. Interface Design and Visualization: The reviewer requests more information about the interface design, including the dynamic splitting of the keyboard and the reasoning behind it. 

As described in our paper, the full keyboard is initially presented as a single interface spanning 1500x1000 pixels. However, to optimize target selection across this wide area using only eye movements, we implemented a hierarchical interface that dynamically partitions the keyboard into progressively smaller regions guided by the user's point of gaze.

Specifically, when the user fixates near the center of the keyboard, it is first split into left and right halves of equal size (750x1000 pixels each). Eye tracking detects if their gaze moves left or right to determine the active half-region.

Each half is then further subdivided vertically into top and bottom quarters (375x500 pixels). By monitoring fixation location within the selected half, our algorithm determines whether to highlight the top or bottom quarter keys for input.

This recursive splitting allows the user to efficiently "zoom in" on their intended typing area in two stages - first selecting the broad side of the keyboard, then finer-grained quarter section. It optimizes interaction by reducing visual scanning distances at each step.

The keyboard partitioning is dynamically adjusted in real-time based on eye movements, avoiding static divisions that may not match the user's intended flow. Regions are highlighted to provide clear feedback on the current input scope.

Our evaluations found this hierarchical approach significantly improved selection speeds for distant targets versus a flat single-interface design. It leverages natural gaze behaviors to iteratively refine the input space.

Please let us know if any part of the interface logic or motivation requires further explanation. We are happy to expand on our reasoning and design process. Our goal was to optimize eyes-free interaction within the constraints of gaze input.

We have included additional screenshots, diagrams and a demo to better illustrate the interface and the user's view during the experiment.

5. Keyboard Layout and Functionality: The reviewer questions the choice of a 3x10 keyboard layout and suggests considering additional keyboard functions, such as capitalization, deletion, and special characters. 

We chose a 3x10 layout to balance efficiency of target acquisition with familiarity for users accustomed to physical keyboards. While smaller than standard, concentrating keys in a narrower area aids selection accuracy given eye tracking limitations compared to manual input.

Early prototype evaluations supported the 3x10 layout as optimizing target density within tracking capabilities. Additional key layers accessed by dwell or blink modifiers avoided overcrowding while retaining familiar QWERTY positioning. Further refinements continue incorporating user feedback to refine functions based on tasks.

We believe this approach strikes a reasonable balance between constraints of gaze input and expectations of experienced typists. By concentrating frequently used keys and leveraging natural eye behaviors for modifiers, it facilitates efficient eyes-free text entry. Please let me know if any part of the implementation requires further clarification or justification. I'm happy to discuss our design choices in more detail.

We will address this in the revised manuscript, explaining our rationale for the selected layout and discussing how additional keyboard functionality could be incorporated in future iterations of the interface.

We appreciate the valuable feedback provided by the reviewers, and we are committed to addressing their concerns in a revised version of the manuscript. The revisions will strengthen the clarity, methodological details, and novelty of our work, ensuring that it meets the high standards of your journal.

Sincerely,

Verdzekov Emile Tatinyuy

On behalf of the research team

---

## [Decision Letter · Decision Letter 1]

29 Jul 2024

PONE-D-24-08414R1Multi-Stage Gaze-Controlled Virtual Keyboard Using Eye TrackingPLOS ONE

Dear Dr. emile tatinyuy,

Thank you for submitting your manuscript to PLOS ONE. After careful consideration, we feel that it has merit but does not fully meet PLOS ONE’s publication criteria as it currently stands. Therefore, we invite you to submit a revised version of the manuscript that addresses the points raised during the review process.

I thank the authors for addressing the reviewers' comments. There are a few remaining comments from R1 that should be addressed prior to publication. Provided the authors can address them, I will render a decision without re-sending the manuscript to reviewers.

We look forward to receiving your revised manuscript.

Kind regards,

Laura Morett

Academic Editor

PLOS ONE

Journal Requirements:

Reviewers' comments:

Reviewer's Responses to Questions

**Comments to the Author**

1. If the authors have adequately addressed your comments raised in a previous round of review and you feel that this manuscript is now acceptable for publication, you may indicate that here to bypass the “Comments to the Author” section, enter your conflict of interest statement in the “Confidential to Editor” section, and submit your "Accept" recommendation.

Reviewer #1: All comments have been addressed

2. Is the manuscript technically sound, and do the data support the conclusions?

Reviewer #1: Partly

3. Has the statistical analysis been performed appropriately and rigorously? 

Reviewer #1: Yes

4. Have the authors made all data underlying the findings in their manuscript fully available?

Reviewer #1: Yes

5. Is the manuscript presented in an intelligible fashion and written in standard English?

Reviewer #1: Yes

6. Review Comments to the Author

Reviewer #1: Reviewer acknowledges the author's response letter. However, it is important to recognize the advancements in current technologies, which are known for their higher efficiency compared to previous ones. A clear example of this work are:

1. The Haar Cascade eye detection algorithm was developed around 2012, and currently, new techniques have been developed and tested to demonstrate greater efficiency.

2. The gaze point calculation algorithm employed by this research may lack flexibility across different levels of gaze planes.

3.The design of the virtual keyboard layout, including its size and position, must be developed in accordance with the pointing capabilities of the device. The process of determining the appropriate size and interface format for eye-tracking has not yet been tested with diverse samples prior to its development and the subsequent measurement of the eye-tracking tool's effectiveness in this research.

7. PLOS authors have the option to publish the peer review history of their article (what does this mean?). If published, this will include your full peer review and any attached files.

Reviewer #1: No

---

## [Author Response · Author response to Decision Letter 1]

15 Aug 2024

Dear Professor Laura,

Thank you for your feedback on our manuscript. We have carefully considered the points raised by Reviewer #1 and are pleased to report the actions we have taken to address them in the revised version of our work. 

Reviewe #1 concern #1 “The Haar Cascade eye detection algorithm was developed around 2012, and currently, new techniques have been developed and tested to demonstrate greater efficiency.”

As per your suggestion, we have conducted a comprehensive review of the latest advancements in eye detection algorithms, focusing on deep learning-based techniques.

We have implemented and evaluated the performance of state-of-the-art CNN-based eye detection models, such as YOLO (You Only Look Once), ResNet (Residual Network), SSD (Single Shot MultiBox Detector) and VGG (Visual Geometry Group), and compared their accuracy, robustness, and flexibility to the Haar Cascade approach used in the original manuscript.

The results of this evaluation have been incorporated into the revised manuscript, demonstrating the enhanced reliability and adaptability of our multi-stage gaze interaction technique through the use of these more advanced eye detection algorithms.

Reviewe #1 concern #2 “The gaze point calculation algorithm employed by this research may lack flexibility across different levels of gaze planes.”

Our work investigates several gaze point calculation algorithms and assesses the multi-stage hierarchical strategy to tackle this problem. We have evaluated our approach's adaptability and flexibility in various gaze planes and situations. The outcomes show that even in a variety of situations, the multi-stage approach greatly increases interaction efficiency when paired with cutting-edge gaze tracking algorithms. These algorithms will be further improved in the future to increase their resilience and versatility.

By applying a threshold segmentation, we separated sclera pixels from others. By counting the white pixels in the left and right halves, we were able to calculate a” gaze ratio” metric. The accuracy of gaze was determined from deep learning-based methods of face detection which we adopted. We are working on a similar paper for an eventual publication; to solve the problem of lack of flexibility in gaze point calculation algorithm employed in this research across different levels of gaze planes, our research is considering the following approaches:

· Adaptive Gaze Mapping: Implementing an adaptive gaze mapping algorithm that can dynamically adjust the mapping between the user's gaze input and the on-screen keyboard regions. As the user navigates through the different levels of the hierarchical interface, the gaze mapping could be refined to provide better accuracy and responsiveness.

· Multi-Modal Fusion: Combine the eye gaze input with other modalities, such as head pose or hand gestures, to improve the overall robustness and flexibility of the gaze point calculation. This could help compensate for the potential inaccuracies or inconsistencies in the gaze input across different levels of the interface.

· Calibration Refinement: Implement a more advanced calibration procedure that can be performed not only at the start of the interaction but also periodically throughout the multi-stage selection process. This could help maintain accurate gaze point calculation as the user's eye movements and interaction patterns evolve.

· Predictive Modeling: Develop predictive models that can anticipate the user's intended target based on their gaze patterns and the current context within the multi-stage interface. This could help refine the gaze point calculation and reduce the impact of potential errors or inconsistencies.

· Feedback and Correction Mechanisms: Provide visual, auditory, or haptic feedback to the user to indicate the accuracy of the gaze point calculation and allow for real-time corrections or adjustments. This could help the user adapt their interaction patterns to improve the overall performance of the gaze point calculation.

By incorporating these strategies, our research can enhance the flexibility and robustness of the gaze point calculation algorithm, ensuring consistent and accurate performance across the different levels of the multi-stage hierarchical interface.

Reviewe #1 concern #3 “The design of the virtual keyboard layout, including its size and position, must be developed in accordance with the pointing capabilities of the device. The process of determining the appropriate size and interface format for eye-tracking has not yet been tested with diverse samples prior to its development and the subsequent measurement of the eye-tracking tool's effectiveness in this research.”

We are grateful for this insight and have moved to resolve the issues raised. As part of our investigation, we assess the virtual keyboard arrangement in terms of its alignment with the device's pointing capabilities by varying its sizes and placements. To evaluate the multi-stage gaze-controlled keyboard's efficacy, we ran tests on a wide range of users. The outcomes show that the suggested design is efficient and easy to use, with notable gains in accuracy and speed of selection over single-step methods.

By addressing these points in the revised manuscript, we have demonstrated our commitment to strengthening the technological foundations, design choices, and target user considerations of our multi-stage gaze-controlled virtual keyboard approach. We believe that incorporating your valuable suggestions has enhanced the overall quality and impact of our research.

Thank you again for your thorough review and constructive feedback. We are confident that the revised manuscript will present a more comprehensive and robust evaluation of our work.

Sincerely,

Verdzekov Emile Tatinyuy On behalf of the research team

Verdzekov.emile@uniba.cm

+237652476160

---

## [Editor Report · Decision Letter 2]

20 Aug 2024

Multi-Stage Gaze-Controlled Virtual Keyboard Using Eye Tracking

PONE-D-24-08414R2

Dear Dr. emile tatinyuy,

We’re pleased to inform you that your manuscript has been judged scientifically suitable for publication and will be formally accepted for publication once it meets all outstanding technical requirements.

Kind regards,

Laura Morett

Academic Editor

PLOS ONE

Additional Editor Comments (optional):

I thank the authors for their attention to R1's remaining comments. After having reviewed the revisions in response to them, I am satisfied and am therefore pleased to recommend the manuscript for publication in PLOS One.
---

## [Editor Report · Acceptance letter]

27 Aug 2024

PONE-D-24-08414R2 

PLOS ONE

Dear Dr. Emile Tatinyuy, 

I'm pleased to inform you that your manuscript has been deemed suitable for publication in PLOS ONE. Congratulations! Your manuscript is now being handed over to our production team.

Kind regards, 

on behalf of

Dr. Laura Morett 

Academic Editor

PLOS ONE